# LEARNING ITERATIVE NEURAL OPTIMIZERS FOR IMAGE STEGANOGRAPHY

**Xiangyu Chen,**[*] **Varsha Kishore**[*]**& Kilian Q Weinberger**
Department of Computer Science
Conell University
Ithaca, NY 14850, USA
{xc429,vk352,kqw4}@cornell.edu

## ABSTRACT

Image steganography is the process of concealing secret information in images through imperceptible changes. Recent work has formulated this task as a classic constrained optimization problem. In this paper, we argue that image steganography is inherently performed on the (elusive) manifold of natural images, and propose an iterative neural network trained to perform the optimization steps. In contrast to classical optimization methods like L-BFGS or projected gradient descent, we train the neural network to also stay close to the manifold of natural images throughout the optimization. We show that our learned neural optimization is faster and more reliable than classical optimization approaches. In comparison to previous state-of-the-art encoder-decoder based steganography methods, it reduces the recovery error rate by multiple orders of magnitude and achieves zero error up to 3 bits per pixel (bpp) without the need for error-correcting codes.

## 1 INTRODUCTION

Image steganography aims to alter a cover image, to imperceptibly hide a (secret) bit string, such that a previously defined decoding method can then extract the message from the altered image. Steganography has been used in many applications such as digital watermarking to establish ownership (Cox et al., 2007), copyright certification (Bilal et al., 2014), anonymized image sharing (Kishore et al., 2021), and for hiding information coupled with images (e.g. patient name and ID in medical CT scans (Srinivasan et al., 2004)). Following the success of neural networks on various tasks, prior work has used end-to-end encoder-decoder networks for steganography (Zhu et al., 2018; Zhang et al., 2019; Hayes & Danezis, 2017a; Baluja, 2017). The encoder takes as input an image $X$ and a message $M$, and produces a steganographic image $\tilde{X}$ that is visually imperceptible from the original image $X$. The decoder recovers the message $M$ from the steganographic image $\tilde{X}$. Such methods can hide large amounts of data in images with great image quality, due to convolutional networks' ability to generate realistic outputs that are close to the input image along the manifold of natural images (Zhu et al., 2016; Zhang et al., 2019). However, these approaches can only reliably encode 2 bits per pixel. At higher bit rates, they suffer from poor recovery error rates for the message $M$ (around 5-25% at 4 bpp). Consequently, they cannot be used for some steganography applications that require the message to be recovered with 100.0% accuracy, for example, if the message is encrypted or hashed. Although error-correcting codes (Crandall, 1998; Munuera, 2007) can be used to recover spurious mistakes, their reliance on additional parity bits reduces the payload, negating much of the advantages of neural approaches.

Recent work has abandoned learned encoders altogether and reformulated image steganography as a constrained optimization problem (Kishore et al., 2021), where the steganographic image is optimized with respect to the outputs of a fixed (random or pre-trained) decoder – by employing a technique based on adversarial image perturbations (Szegedy et al., 2013). The optimization problem is solved with off-the-shelve gradient-based optimizers, such as projected gradient descent (Carlini & Wagner, 2017) or L-BFGS (Fletcher, 2013). Such approaches achieve low error rates with high payloads (2-3% error at 4 bpp), but are slow and prone to getting stuck in local minima. Further,

---

[*]Equal contribution.

each pixel is optimized in isolation, and pixel-level constraints only ensure that the steganographic image stays close to the input image according to an algebraic norm, rather than along the natural image manifold. Although similar manifold-unaware approaches are successfully deployed for adversarial attacks, steganography aims to precisely control millions of binary decoder outputs, instead of a single class prediction; the resulting optimization problem is thus harder and prone to producing unnatural-looking images.

In this paper we introduce *Learned Iterative Steganography Optimization (LISO)*, a method that combines the ability of neural networks to learn image manifolds with the rigor of constrained optimization. LISO is highly efficient in hiding messages inside images, and achieves very low message recovery error. LISO achieves these feats with an encoder-decoder-critic architecture. The encoder is a *learned* recurrent neural network that iteratively solves (Teed & Deng, 2020; Alaluf et al., 2021) the constraint optimization problem proposed in Kishore et al. (2021) and mimics the steps of a gradient-based optimization method. Figure 1 demonstrates how the steganographic image and error rate change over subsequent iterations. We also use a critic network (Zhang et al., 2019) to ensure the changes stay imperceptible and that the steganographic image looks natural. Our resulting architecture can be trained end-to-end.We show that LISO learns a more efficient descent direction than standard (manifold unaware) optimization al-

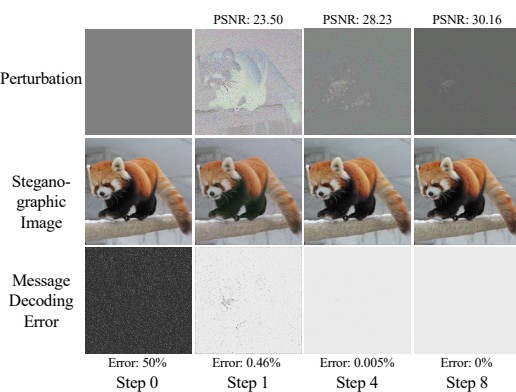

Figure 1: LISO iteratively optimizing a sample image with 3 bits encoded in each pixel. The recovery error goes down drastically to exactly zero in only 8 steps. The perturbation (difference between the steganographic image and the original image) is large in the beginning, but becomes imperceptible with LISO's optimization, leading to a natural-looking steganographic image.

gorithms and produces better steganography results with great consistency. Finally, the error rate can (almost) always be driven to a flat 0 if the optimization is finished with a few iterations of L-BFGS within the vicinity of LISO's solution (LISO + L-BFGS).

We evaluate the efficacy of LISO extensively across multiple datasets. We demonstrate that at test-time, with unseen cover images and random bit strings, the optimizer can reliably circumvent bad local minima and find a low-error solution within only a few iterative steps that already outperforms all previous encoder-decoder-based approaches. If the optimization is followed by a few additional updates of L-BFGS optimization, we can reliably reach 100% error-free recovery even with 3 bits per pixel (bpp) hidden information across all (thousands of) images we tested on.

Our contributions are as follows. 1) We introduce a novel gradient-based neural optimization algorithm, LISO, that learns preferred descent directions and is image manifold aware. 2) We show that its optimization process based on learned descent directions is *orders of magnitudes* faster than classical optimization procedures. 3) As far as we know, our variant LISO + L-BFGS is by far the most accurate steganography algorithm in existence, resulting in perfect recovery on *all* images we tried with 3bpp and the vast majority for 4bpp — implying that it avoids bad local minima and can be deployed without error correcting code. The code for LISO is available at https://github.com/cxy1997/LISO.

## 2 RELATED WORK

**Steganography.** Classic steganography methods operate directly on the spatial image domain to encode data. Least Significant Bit (LSB) Steganography encodes data by replacing the least significant bits of the input image pixels with bits from the secret data sequentially (Chan & Cheng, 2004). Pixel-value Differencing (PVD) (Wu & Tsai, 2003) hides data by comparing the differences between the intensity values of two successive pixels. Following this, many methods were presented to hide data while minimizing distortion to the input images and these methods differed in how the distortion was measured. Highly Undetectable Steganography (HUGO) (Pevnỳ et al., 2010) identifies

pixels that can be modified without resulting in a large distortion based on handcrafted steganalytic features. Wavelet Obtained Weights (WOW) (Holub & Fridrich, 2012) uses syndrome-trellis codes to minimize the expected distortion and to identify noisy or high-textured regions to modify.

With the advent of deep learning, many deep learning approaches have been proposed for steganography. These approaches either use neural networks in some stages of the pipeline (e.g., as a feature extractor) or learn networks end-to-end for message encoding and decoding. HiDDeN (Zhu et al., 2018) proposed an end-to-end encoder-decoder architecture to hide up to 0.4 bpp of information in images. Hayes & Danezis (2017b) uses adversarial training to approximately hide 0.4 bpp in $32 \times 32$ images. SteganoGAN Zhang et al. (2019) adopts adversarial training to generate steganographic images with better quality, and can hide up to 6 bpp with error rates of about 13-33%. Tan et al. (2021) improved upon SteganoGAN and introduce CHAT-GAN which uses an additional channel attention module to identify favorable channels in feature maps to hide bits of information.

HiDDeN, SteganoGAN, and CHAT-GAN hide arbitrary bit strings but other methods hide a (structured) image within an image (Baluja, 2017; Dong et al., 2018; Yu, 2020; Rahim et al., 2018) by learning image priors and condensing information; arbitrary bit strings can not be similarly condensed. Invertible Steganography Network (Lu et al., 2021) hides multiple images in a single cover image using an invertible neural network. Deep Multi-Image Steganography with Private Keys (Shang et al., 2020) hides multiple images in a cover image while producing many secret keys, each of which is used to extract exactly one of the hidden images. HiNet (Jing et al., 2021) is an invertible network to hide information in the wavelet domain. Zhang et al. (2020) explore finding cover-agnostic universal perturbations that can be added to any natural image to hide a secret image. There have also been attempts to conduct steganography with non-digital images (Tancik et al., 2020; Wengrowski & Dana, 2019), but in this paper we focus on digital images.

**Steganalysis.** Many applications require that steganographic images are indistinguishable from natural images. Steganalysis systems are used to detect whether secret information is hidden in images and they can broadly be divided into two categories – statistical and neural. The former makes use of statistical tests on pixels to determine whether a message is encoded in an image (Dumitrescu et al., 2002b; Fridrich et al., 2001; Westfeld & Pfitzmann, 1999; Dumitrescu et al., 2002a), while the latter employs neural networks to detect the existence of hidden messages. Classic steganography methods can be easily detected by statistical tools, but more recent methods are complex and harder to detect with statistical tools and require neural tools. Neural steganalysis methods have shown state-of-the-art performances for detecting steganography with high accuracy even when low payloads of information are hidden. GNCNN (Qian et al., 2016) used handcrafted features and CNNs to perform steganalysis. Following that, many methods have used deep networks to automatically extract features and perform steganalysis (You et al., 2020; Qian et al., 2015; Chen et al., 2017; Wu et al., 2019; Qian et al., 2018). In response to the advances in steganalysis, neural steganography methods add steganalysis systems into their end-to-end pipeline in order to find robust perturbations that evade detection by steganalysis systems (Shang et al., 2020; Dong et al., 2018).

**Learning to Optimize.** Recent methods have investigated incorporating optimization problems into neural network architectures (Amos & Kolter, 2017; Agrawal et al., 2019; Wang et al., 2019). These methods design custom neural network layers that mimic certain canonical optimization problems. As a result, parametrized optimization problems can be inserted into neural networks. Not all optimization algorithms can easily be written as a layer and another strategy is to train a network to learn iterative updates from data (Adler & Öktem, 2018; Flynn et al., 2019; Gregor & LeCun, 2010). Most similar to our approach is arguably RAFT (Teed & Deng, 2020), which poses optical flow estimation as an optimization problem and uses a recurrent neural network to find an optimal descent direction. In a similar vein, we aim to learn the steganography optimization problem with a convolutional neural network without specially designed optimization layers.

## 3 NOTATION AND SETUP

Let $\boldsymbol{X} \in [0,1]^{H \times W \times 3}$ denote an RGB cover image with height $H$ and width $W$.[1] The hidden message $\boldsymbol{M}$ is a bit string of length $m = H \times W \times B$ that is reshaped to match the cover image's size,

---

[1]In practice, we use PNG-24/JPEG with quantized pixel values.

i.e. $M \in \{0, 1\}^{H \times W \times B}$, where payload $B$ is the number of encoded bits per pixel (bpp).[2] Image steganography aims to conceal $M$ in a steganographic image $\tilde{X} \in [0, 1]^{H \times W \times 3}$ that looks visually identical to $X$, such that $M$ is transmitted undetected through $\tilde{X}$. A steganographic encoder *Enc* takes as input $X$ and $M$ and outputs $\tilde{X}$. A decoder *Dec* recovers the message, $M' = Dec(\tilde{X})$, with minimal recovery error $\epsilon = \frac{\|M' - M\|_0}{m}$ (ideally $\epsilon = 0$). Below we describe two recent paradigms for steganography ( Appendix F summarizes the similarities and differences between the paradigms).

**Learned Image Steganography.** Recent deep-learning-based image steganography methods (Zhang et al., 2019; Zhu et al., 2018; Tan et al., 2021) use straight-forward convolutional neural network architectures for encoders and decoders that maintain the same feature dimensions, $H \times W$, throughout (i.e. no downsampling). SteganoGAN (Zhang et al., 2019) and CHAT-GAN (Tan et al., 2021) use such an encoder network to generate the steganographic image $\tilde{X}$ with a single forward pass and a decoder network to predict the message $M'$. The final layer of the decoder usually consists of $H \times W \times B$ sigmoidal outputs, producing real values within $[0, 1]$ that turn into the binary recovered message $M'$ after rounding. These methods also have a $Critic/Discriminator$ to predict whether an image looks natural. The encoder, decoder, and critic are all trained end-to-end.

**Optimization-based Image Steganography.** Given a differentiable decoder with random or learned weights (from a method mentioned in the last paragraph), Kishore et al. (2021) express the steganography encoding step as a per sample optimization problem that is inspired by adversarial perturbations (Szegedy et al., 2013). Their method, Fixed Neural Network Steganography (FNNS), finds a steganographic image by solving the following constrained optimization problem while ensuring that the perturbed image lies within the $[0, 1]^{H \times W \times 3}$ hypercube:

$$\min_{\tilde{X} \in [0,1]^{H \times W \times 3}} L_{\text{acc}}\left(Dec\left(\tilde{X}\right), M\right) + \lambda L_{\text{qua}}\left(\tilde{X}, X\right)$$

$$L_{\text{acc}}\left(M', M\right) := \langle M, \log M' \rangle + \langle (1 - M), \log (1 - M') \rangle$$

$$L_{\text{qua}}\left(\tilde{X}, X\right) := \frac{1}{N}\|\tilde{X} - X\|_2^2, \tag{1}$$

where $\langle \cdot \rangle$ denotes the dot product operation, $\lambda$ is a weight factor and $N = H \times W \times 3$. Note that only the perturbed image $\tilde{X}$ is being optimized and the decoder *Dec* is kept fixed throughout. The accuracy loss $L_{\text{acc}}\left(M', M\right)$ is a binary cross entropy loss that encourages the message recovery error to be low, and the quality loss $L_{\text{qua}}\left(X, \tilde{X}\right)$ uses mean squared error to ensure that the steganographic image looks similar to the cover image. We denote the objective in Equation 1 as $\ell(\tilde{X}, M)$. Several solvers can be used to solve Equation 1 as shown in Algorithm 1, but the most popular choices are iterative, gradient-based hill-climbing algorithms. In the algorithm, $\eta > 0$ denotes the step size, and $g(\cdot)$ is an update function defined by the specific optimization method. The perturbation $\delta$ is iteratively optimized to minimize the loss $\ell$ within the image pixel constraints.

**Drawbacks of Existing Methods** Existing learned methods are fast because single forward passes with the trained encoder and decoder are all that are needed for encoding and decoding, but they yield high error rates of up to 5-25% for 4 bpp. It is also hard to incorporate additional constraints to these models without retraining. On the other hand, optimization-based approaches can result in lower error rates, but they are much slower (requiring thousands of iterations). Furthermore, their resulting recovery error depends heavily on the weights of the decoder and the initialization of $\delta$ (Kishore et al., 2021). Especially with $B \geq 4$, the optimization can get stuck in local minima.

## 4    LEARNED STEGANOGRAPHIC OPTIMIZATION

Ideally, we would like a method that is both **fast** and has **low recovery error**, so we combine ideas from learned methods and iterative optimization methods to propose *Learned Iterative Steganography Optimization (LISO)*. LISO consists of an encoder, a decoder, and a critic network, but the encoder is iterative and it approximates the function $g(\cdot)$ in Algorithm 1. The optimization procedure can be viewed as a recurrent neural network with a hidden state that iteratively optimizes the perturbation $\delta$ with respect to the loss $\ell$. At each iteration, it obtains the previous estimate $\delta_{t-1}$ and the gradient of $\ell$ as input and produces a new $\delta_t$. The hidden state allows LISO to learn what

---

[2] If $m$ is not a multiple of $H \times W$, it is padded with zeros.

optimization steps are best suited for this task. After multiple iterative steps, the encoder outputs the steganographic image. The decoder is a simple feed-forward convolutional neural network trained to retrieve the message from the steganographic image. The critic is a neural classifier (akin to GAN discriminators) trained to discriminate between cover images and steganographic images (Hayes & Danezis, 2017b). As everything is differentiable, the iterative encoder, the decoder, and the critic can be jointly learned on a dataset of images.

**LISO architecture.** The LISO decoder and critic have 3 convolutional blocks, each containing a convolutional layer (with $3\times3$ kernels and stride 1), batch-norm, and leaky-ReLU, similar to Zhang et al. (2019). In the decoder, the convolutional blocks are followed by another convolutional layer to obtain a $H\times W\times B$ dimensional output. In the critic, the convolutional blocks are followed by an adaptive mean pooling layer to obtain a scalar output that denotes a prediction of whether an image has a hidden code.

The core of LISO is its encoder, which has a recurrent GRU-based update operator that functions like Algorithm 1. The function $g(\cdot)$ in Algorithm 1 is approximated using a fully convolutional network designed around a gated recurrent unit (GRU) (Cho et al., 2014). The GRU has an internal state and keeps track of information relevant to the optimization (more details about the GRU cell are in Appendix A). To avoid information loss, the LISO networks have no down-sampling/up-sampling layers, which means LISO can process images of any size. At a high level, the LISO encoder (illustrated

---

**Algorithm 1** Iterative Optimization

$\boldsymbol{\delta}_0 \leftarrow \mathbf{0}$
**for** t=1 to T **do**
$\qquad \boldsymbol{\delta}_t \leftarrow \boldsymbol{\delta}_{t-1} +$
$\qquad\qquad \eta \cdot g\left(\nabla_{\boldsymbol{\delta}_{t-1}} \ell\left(\boldsymbol{X}+\boldsymbol{\delta}_{t-1}, \boldsymbol{M}\right), \boldsymbol{X}, \boldsymbol{\delta}_{t-1}\right)$
$\tilde{\boldsymbol{X}} \leftarrow \boldsymbol{X}+\boldsymbol{\delta}_T$

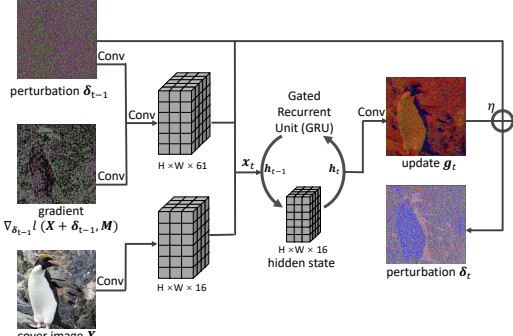

Figure 2: Model architecture of LISO's iterative optimization network that functions as optimizer $g$ in Algorithm 1.

---

in Figure 2) takes as input the cover image $\boldsymbol{X}$, the current perturbation $\boldsymbol{\delta}_{t-1}$ and the gradient of the loss with respect to the perturbed image $\nabla_{\boldsymbol{\delta}_{t-1}} l(\boldsymbol{X}+\boldsymbol{\delta}_{t-1}, \boldsymbol{M})$ and predicts the perturbation $\boldsymbol{\delta}_t$ for the next step. The GRU hidden state $\boldsymbol{h}_0$ is initialized with a feature extractor (not shown) that is a simple 3-layer fully convolutional network that extracts features from the input image. The input to the GRU in subsequent steps $x_t$ is constructed by concatenating the extracted features from current perturbation $\boldsymbol{\delta}_{t-1}$, the gradient $\nabla_{\boldsymbol{\delta}_{t-1}} l(\boldsymbol{X}+\boldsymbol{\delta}_{t-1}, \boldsymbol{M})$ and the cover image $\boldsymbol{X}$. The GRU unit updates its own hidden state $h_t$, and the hidden state is processed by additional convolutional layers to produce a gradient-type step update $\boldsymbol{g}_t$. The final update becomes $\boldsymbol{\delta}_t = \boldsymbol{\delta}_{t-1} + \eta \boldsymbol{g_t}$, where $\eta$ is the step size for the new update. Perturbation $\boldsymbol{\delta}_t$ can also be clipped by a truncation factor to limit how much the $\boldsymbol{X}$ changes. The steganographic image, that is the final output of the LISO encoder, is produced through recurrent applications of the iterative encoder; the final steganographic image becomes $\tilde{\boldsymbol{X}} = \boldsymbol{X} + \eta \sum_t \boldsymbol{g}_t$.

**Training.** The entire LISO pipeline (the iterative encoder, decoder, and critic) is trained end-to-end on a dataset of images. As with GAN training, we optimize the critic and the encoder-decoder networks in alternating steps. During the training, we compute loss for all intermediate updates with exponentially increasing weights ($\gamma^{T-t}$ at step $t$). With intermediate predictions denoted as $\tilde{\boldsymbol{X}}_1, \ldots, \tilde{\boldsymbol{X}}_T$ the loss is:

$$L_{train} = \sum_{t=1}^{T} \gamma^{T-t} \big[ L_{acc}\left(\boldsymbol{M}, \tilde{\boldsymbol{X}}_t\right) + \lambda L_{qua}\left(\boldsymbol{X}, \tilde{\boldsymbol{X}}_t\right) + \mu L_{crit}\left(\boldsymbol{X}, \tilde{\boldsymbol{X}}_t\right) \big], \qquad (2)$$

where $\gamma \in (0, 1)$ is a decay factor and $L_{crit}$ denotes the critic loss (with weight $\mu > 0$). Despite the predictions being sub-optimal from earlier iterations, we still make use of the loss from those steps because they do provide useful update directions for later iterations. Note that this loss is similar to Equation 1, but there is an additional critic loss term that ensures that the steganographic image looks like a natural image (Zhang et al., 2019). As the qualitative and critic losses are computed at all steps, the optimizer is encouraged to stay on or near the manifold of natural images throughout.

Table 1: Image steganography results of LISO on three different datasets with different payloads.

| | Method | Error Rate (%) ↓ | | | | PSNR ↑ | | | | SSIM ↑ | | | |
|---|---|---|---|---|---|---|---|---|---|---|---|---|---|
| | | 1 bit | 2 bits | 3 bits | 4 bits | 1 bit | 2 bits | 3 bits | 4 bits | 1 bit | 2 bits | 3 bits | 4 bits |
| Div2k | SteganoGAN | 5.12 | 8.31 | 13.74 | 22.85 | 21.33 | 21.06 | 21.42 | 21.84 | 0.76 | 0.76 | 0.77 | 0.78 |
| | FNNS-D | **0.00** | **0.00** | 0.10 | 5.45 | 29.30 | 26.25 | 22.90 | 25.74 | 0.82 | 0.73 | 0.53 | 0.65 |
| | LISO | 4E-05 | 4E-04 | 2E-03 | 3E-02 | 33.83 | 33.18 | 30.24 | 27.37 | 0.90 | 0.90 | 0.85 | 0.76 |
| | LISO+L-BFGS | **0.00** | **0.00** | **0.00** | 1E-05 | 33.12 | 32.77 | 29.58 | 27.14 | 0.89 | 0.89 | 0.82 | 0.74 |
| CelebA | SteganoGAN | 3.94 | 7.36 | 8.84 | 10.00 | 25.98 | 25.53 | 25.70 | 25.08 | 0.85 | 0.86 | 0.85 | 0.82 |
| | FNNS-D | **0.00** | **0.00** | **0.00** | 3.17 | 36.06 | 34.43 | 30.05 | 33.92 | 0.87 | 0.86 | 0.71 | 0.84 |
| | LISO | 4E-04 | 1E-03 | 4E-03 | 1E-01 | 35.62 | 36.02 | 32.25 | 30.07 | 0.89 | 0.90 | 0.82 | 0.79 |
| | LISO+L-BFGS | **0.00** | **0.00** | **0.00** | 7E-05 | 35.40 | 35.63 | 31.61 | 28.36 | 0.89 | 0.89 | 0.79 | 0.71 |
| MS COCO | SteganoGAN | 3.40 | 6.29 | 11.13 | 15.70 | 25.32 | 24.27 | 25.01 | 24.94 | 0.84 | 0.82 | 0.82 | 0.82 |
| | FNNS-D | **0.00** | **0.00** | **0.00** | 13.65 | 37.94 | 34.51 | 27.77 | 34.78 | 0.95 | 0.90 | 0.72 | 0.89 |
| | LISO | 2E-04 | 5E-04 | 3E-03 | 4E-02 | 33.83 | 32.70 | 27.98 | 25.46 | 0.90 | 0.89 | 0.75 | 0.68 |
| | LISO+L-BFGS | **0.00** | **0.00** | **0.00** | 1E-05 | 33.42 | 31.80 | 26.98 | 25.17 | 0.90 | 0.87 | 0.70 | 0.67 |

**Inference.** The learned LISO iterative encoder and decoder are used during inference. The iterative encoder performs a gradient-style optimization on the cover image to produce the steganographic image with the desired hidden message. Mathematical optimization algorithms like L-BFGS have strong theoretical convergence guarantees, which assure fast and highly accurate convergence within the vicinity of a global minimum, but for non-convex optimization problems they have a tendency to get stuck in local minima. Although our learned optimizer lacks theoretical convergence guarantees, LISO is very efficient at finding local vicinity of good minima for natural images—it can reliably find solutions with only a few bits decoded incorrectly (on the order of 0.01%) as shown in Table 1. FNNS (Kishore et al., 2021) works in a complementary manner to trained encoder-decoder networks and optimizes steganographic images using fixed networks and off-the-shelf optimization algorithms. Consequently, we can use LISO in conjunction with FNNS; we refer to this procedure as LISO+(PGD) or LISO+(LBFGS), dependent on which optimization algorithm is used for FNNS optimization. Concretely, we encode a message into a cover image by using the LISO encoder, and then use L-BFGS/PGD to further reduce the error rate. The error rate after LISO is already quite low and hence only a few additional L-BFGS iterations are required to reduce the error. Compared to FNNS, LISO can reliably achieve 0% error for up to 3 bpp, with drastically fewer iterations.

## 5 EXPERIMENTS AND DISCUSSION

We evaluate our method on three public datasets: 1) Div2k (Agustsson & Timofte, 2017) which is a scenic images dataset, 2) CelebA (Liu et al., 2018) which consists of facial images of celebrities, and 3) MS COCO (Lin et al., 2014) which contains images of common household objects and scenes. For CelebA and MS COCO we use the first 1000 for validation and the following 1,000 for training. Note that 1,000 training images are sufficient for LISO to achieve SOTA performance, and using full datasets will only make training longer. The messages are random binary bit strings sampled from an independent Bernoulli distribution with $p = 0.5$, which is close to the distribution of compressed/encrypted messages.

During training, we set the number of encoder iterations $T = 15$, the step size $\eta = 1$, the decay $\gamma = 0.8$ and loss weights $\lambda = \mu = 1$. During inference, we use a smaller step size $\eta = 0.1$ for a larger number of iterations $T$; we iterate until the error rate converges. We show the average number of iterations required to minimize the recovery error in Table 3. The average number is consistently below the number of iterations used during training $T = 15$, although some images require up to 25 iterations. These results suggest that in practice LISO just needs a few iterations to achieve low recovery error rates. Figure 3 shows how the recovery error decreases over time during testing. The monotonicity of the graphs indicates that the optimization loss is well aligned with recovery error and that LISO is successful at learning to find suitable descent directions (more details are in the subsequent optimization analysis). The final steganographic images are stored in PNG format.

**Image Steganography Performance.** Most traditional steganography methods hide $< 1$ bpp messages in images, and methods that hide large payloads of information usually hide structured information like images. Very few methods are designed for hiding **high payload arbitrary bit string messages** in color images. We compare LISO with three other methods that are designed to do so, SteganoGAN (Zhang et al., 2019), FNNS (Kishore et al., 2021) and CHAT-GAN (Tan et al., 2021).

Since the source code for CHAT-GAN was not available, we were only able to compare against the numbers provided in the paper on MS-COCO and these results are presented in Appendix E. In Table 1, we evaluate LISO's performance on image steganography in terms of both message recovery accuracy and image quality. Following previous work (Zhang et al., 2019), we measure how much the steganographic image changes by computing the structural similarity index (SSIM) and peak signal-to-noise ratio (PSNR) (Wang et al., 2004) between the cover and steganographic images. In Appendix B, we also evaluate with the Learned Perceptual Image Patch Similarity (LPIPS) metric (Zhang et al., 2018) and find trends similar to that of PSNR and SSIM.

From Table 1, we observe that the error rates of LISO are an order of magnitude lower when compared to SteganoGAN. Interestingly, the image quality as measured by PSNR scores is also superior. LISO is able to achieve an error rate of **exactly 0%** for payloads $\leq 3$ bpp for all images when it is paired with a few steps of direct L-BFGS optimization (the numbers in Table 1 are averaged and for many samples 0% error is obtained just with LISO). Using a learned iterative method, we are able to achieve the same performance as that obtained from a direct optimization method (FNNS), but in far fewer optimization steps and time (see Table 4). Essentially LISO's ability to learn from past optimizations and to stay on the image manifold, allows it to achieve low error rates, high image quality, and initializations (for further L-BFGS optimization) near a good minimum.

Figure 4 shows the steganographic images produced by LISO under different payloads. Qualitatively we observe that the image quality is quite good even when a large number of bits per pixel are hidden in the cover images. The CelebA example shows a rare example where the hue changes slightly while the image remains un-pixelated and natural; in contrast, the low-quality steganographic images produced by FNNS tend to be pixelated. Different from previous image steganography methods that produce unnatural Gaussian noise, LISO's perturbation is smooth and natural because LISO implicitly learns the manifold of natural images and produces steganographic images that stay close it. These color shifts disappear when a larger image quality factor $\lambda$ or a more diverse dataset is used (a similar observation was made by Upchurch et al. (2017) for image transformations). We can also explicitly control the trade-off between error rate and image quality by changing the image quality factors $\lambda$ and $\mu$ on the loss terms if desired. Additional qualitative images obtained by varying these are provided in Appendix J.

**Optimization with LISO.** In Figure 3, we compare optimizing with the LISO encoder against popular numerical optimization algorithms including projected gradient descent (Madry et al., 2017) and L-BFGS (Liu & Nocedal, 1989) on the steganographic encoding task. Note that *L-BFGS-Random* is equivalent to *FNNS-R* and *L-BFGS-Pretrained* is equivalent to *FNNS-D* (Kishore et al., 2021). LISO converges faster than any other optimization algorithm and this validates our hypothesis that LISO can learn a descent direction that is better than the one found by common gradient-based optimization methods; LISO converges to $100\times$ lower loss and $80\times$ lower error rate. This is because the LISO encoder can leverage information learned from optimizing many

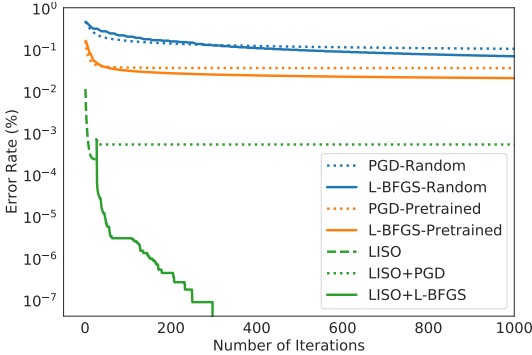

Figure 3: Error-iteration curves for different optimization methods (at 4 bpp). The pretrained network is SteganoGAN.

samples, while off-the-shelf gradient-based methods perform per-sample optimization. Furthermore, we show that recovery error rate from LISO can further be reduced by taking only a few additional *L-BFGS* steps. This suggests that the LISO encoder output is quite close to the optimum and is a good initialization for using conventional optimization methods like *L-BFGS* or *PGD*.

**Steganography with JPEG Compression.** JPEG (Wallace, 1992) is a commonly used lossy image compression method, which removes high-frequency components for reduced file size. JPEG-compression-resistant steganography is hard because steganography and JPEG compression have opposing objectives; steganography attempts to encode information through small imperceptible changes and JPEG compression attempts to eliminate texture details. General-purpose steganography methods have about 50% recovery error rate when decoding JPEG-compressed images. Many methods have been specifically developed to perform JPEG-resistant steganography to hide $< 0.5$

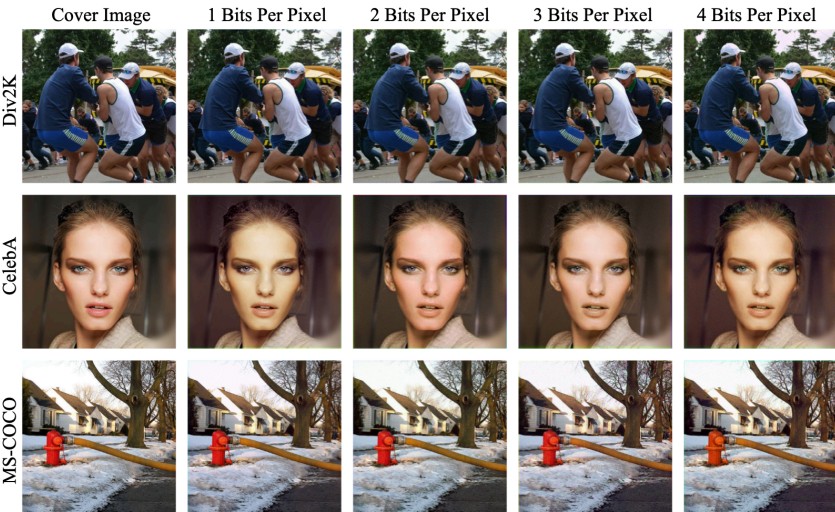

Figure 4: Random images with corresponding steganographic images under different payloads.

bpp messages (Zhang et al., 2015; Fan et al., 2011; Zhang et al., 2016; Holub et al., 2014). Some of these find perturbations specifically in low-frequency space to avoid being removed by compression. HiDDeN (Zhu et al., 2018) and FNNS (Kishore et al., 2021) include differentiable JPEG layers to approximate compression as part of their network architecture. Following (Athalye et al., 2018), we adopt an approximate JPEG layer where the forward pass performs normal JPEG compression and the backward pass is an identity function. We call this adapted method LISO-JPEG. Table 2 compares LISO-JPEG with existing methods. LISO is able to find perturbations that won't be lost in JPEG compression, while maintaining low error rate (<1E-01). The PSNR values for LISO-JPEG are low but qualitative 1bpp steganographic image examples from LISO-JPEG (see Figure 5) show that there is only a slight pixelation. As seen in Table 2, the PSNR of FNNS-JPEG and LISO-JPEG are similar but LISO-JPEG has a significantly lower error rate. Higher quality steganographic images can be obtained by training a LISO model with a higher weight on the qualitative loss $L_{qua}$, but this will cause a slight increase in the error rate.

Table 2: LISO-JPEG results compared with baselines. JPEG quality 80 is used.

| Method | bpp | Error (%) ↓ | PSNR ↑ |
|--------|-----|-------------|--------|
| LISO-PNG | 1 | 2E-05 | 31.99 |
| FNNS-JPEG | 1 | 32.03 | 22.65 |
| HiDDeN | 0.2 | ≥10 | unknown |
| LISO-JPEG | 1 | 6E-02 | 19.72 |

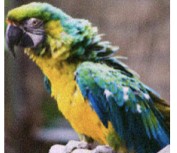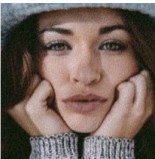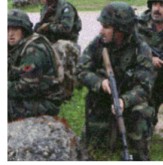

Figure 5: LISO-JPEG steganographic images.

**Computational Time.** One of the advantages of using a learned method as opposed to an optimization method is that it is much faster. The average inference time required for the different learned and optimization-based methods is presented in Table 4. The fastest method is SteganoGAN because all that is required to use SteganoGAN is single forward passes. The FNNS variants, on the other hand, are much slower. Our proposed method LISO, is slightly slower than SteganoGAN but not by a significant magnitude. The reported times were the average times on Div2k's validation set and the methods were run on a Nvidia Titan RTX GPU.

**Steganalysis.** To investigate the robustness of our method against statistical detection techniques, we use Stegexpose (Boehm, 2014), a tool devised for steganography detection using four well-known steganalysis approaches and find that the detection rate is lower than 25% for images produced by LISO. Recently, more powerful neural-based methods are being used to detect steganography (see related work). It is hard for any steganography method, including classical methods like WOW and S-UNIWARD, to evade detection from these neural methods even with low payload messages of about 0.4bpp (You et al., 2020). However, we show that in the following two scenarios we can evade detection by SiaStegNet (You et al., 2020).

Table 3: Number of iterations it takes to reach optimum under payload of 1-4 bits per pixel on Div2k's validation set.

| Bits per Pixel | Iterations (avg) | Max Iterations |
|---|---|---|
| 1 | 6.44 | 18 |
| 2 | 9.50 | 17 |
| 3 | 11.36 | 21 |
| 4 | 11.15 | 24 |

Table 4: Average computation time (in seconds). (∗ indicates the optimization was stopped after zero error was reached.)

| Method | 1 bit | 2 bits | 3 bits | 4 bits |
|---|---|---|---|---|
| SteganoGAN | 0.09 | 0.09 | 0.08 | 0.09 |
| FNNS-R | 42.18 | 114.44 | 156.91 | 159.19 |
| FNNS-D | 4.95* | 10.53* | 44.39 | 44.29 |
| LISO | 0.16 | 0.29 | 0.32 | 0.33 |
| LISO+LBFGS | 1.26* | 2.81* | 6.64* | 28.59 |

Table 5: Steganalysis results using images produced by different methods, evaluated on MS-COCO dataset. SG is used to denote SteganoGAN.

| Setting | Method | Error Rate (%) ↓ | | | | PSNR ↑ | | | | Detection Accuracy Rate (%) ↓ | | | |
|---|---|---|---|---|---|---|---|---|---|---|---|---|---|
| | | 1 bit | 2 bits | 3 bits | 4 bits | 1 bit | 2 bits | 3 bits | 4 bits | 1 bit | 2 bits | 3 bits | 4 bits |
| w/o defense | SG | 3.40 | 6.29 | 11.13 | 15.7 | 25.32 | 24.27 | 25.01 | 24.94 | 52 | 57 | 92 | 94 |
| | FNNS-D | 0.00 | 0.00 | 0.01 | 13.65 | 37.94 | 34.51 | 27.77 | 34.78 | 73 | 75 | 99 | 98 |
| | LISO | 1E-04 | 3E-04 | 5E-03 | 4E-02 | 33.83 | 34.15 | 30.08 | 24.58 | 51 | 42 | 98 | 87 |
| w/ defense | FNNS-D | 0 | 0.00 | 0.00 | 0.01 | 22.28 | 22.35 | 21.2 | 21.36 | 7 | 12 | 14 | 83 |
| | LISO | 4E-03 | 1E-02 | 8E-03 | 5E-02 | 31.62 | 28.67 | 26.63 | 25.10 | 4 | 8 | 5 | 5 |

In the first scenario, "w/o defense", the steganography model $M$ is trained without any explicit precautions to avoid steganalysis detection. We assume the attacker (i.e. the party applying steganalysis) knows the model architecture of $M$ but has no access to the actual weights, exact training set, or hyperparameters. However, they can train a surrogate model $M'$ in order to create their own training set of steganographic images. To simulate this setting, we used a steganalysis model trained on CelebA to detect steganographic MS COCO images. From Table 5 we see that in the "w/o defense" scenario, steganographic images from SteganoGAN and FNNS can be detected much more easily when compared to images from LISO.

In the second scenario, "w/ defense", we take advantage of the fact that neural steganalysis methods are fully differentiable, and both LISO and FNNS involve gradient-based optimization. It is therefore possible to avoid detection by adding an additional loss term from the steganalysis system into the LISO optimization or the FNNS optimization. Specifically, during evaluation we add the logit value of the steganographic class to the loss if an image is detected as steganographic. Note that we cannot add an additional loss term to SteganoGAN during inference because SteganoGAN just uses forward passes of a trained neural network for inference. We also tried training a SteganoGAN network with an auxiliary steganalysis loss and found that we couldn't reliably avoid detection even when we did so. As seen in Table 5, we obtain single-digit detection accuracy rates for LISO in the "w/ defense" scenario for all bit rates. Compared to FNNS, LISO is able to avoid detection more effectively. The image quality of steganographic images produced by LISO is also better than those produced by FNNS under this scenario. We also tested LISO with two additional steganalysis systems, SRNet (Boroumand et al., 2018) and XuNet (Xu et al., 2016), and the corresponding results are presented in Appendix H.

# 6 CONCLUSION

We propose a novel iterative encoder-based method, LISO, for image steganography. The LISO encoder is designed to learn the update rule of a general gradient-based optimization algorithm for steganography, and is trained simultaneously with a corresponding decoder. It learns a more efficient dynamic update rule for steganography when compared to PGD or L-BFGS, and the learned decoder is particularly suitable for further L-BFGS optimization. LISO also implicitly learns the manifold of images and therefore produces high-quality steganographic images. LISO achieves state-of-the-art results for steganography, while being fairly fast. It is also flexible and can incorporate additional constraints, like producing JPEG-resistant steganographic images or making steganographic images undetectable by specific steganalysis systems. In the future, we plan to investigate ways to increase image quality while maintaining low error for JPEG-resistant steganography.

ACKNOWLEDGEMENTS

This research is supported by grants from DARPA AIE program, Geometries of Learning (HR00112290078), DARPA Techniques for Machine Vision Disruption grant (HR00112090091), the National Science Foundation NSF (IIS-2107161, III1526012, IIS-1149882, and IIS-1724282), and the Cornell Center for Materials Research with funding from the NSF MRSEC program (DMR-1719875). We would like to thank Oliver Richardson and all the reviewers for their feedback.

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

# A GRU Operations

$$z_t = \sigma\left(\text{Conv}_{3\times3}\left([h_{t-1}, x_t], W_z\right)\right)$$
$$r_t = \sigma\left(\text{Conv}_{3\times3}\left([h_{t-1}, x_t], W_r\right)\right)$$
$$\bar{h}_{t-1} = \tanh\left(\text{Conv}_{3\times3}\left([r_t \odot h_{t-1}, x_t], W_h\right)\right)$$
$$h_t = (1 - z_t) \odot h_{t-1} + z_t \odot \bar{h}_{t-1} \tag{3}$$

As described in the paper, the Gated Recurrent Unit (GRU) is an integral part of the LISO encoder. The operations of the GRU used in LISO are defined in Equation 3, where $x_t$ is the input to the GRU in $t^{\text{th}}$ iteration, $h_t$ is the GRU's hidden state, and $W_z, W_r, W_h$ are GRU's weight matrices. For LISO, $x_t$ is the concatenation of features extracted from the input image, the gradient from the loss, and the perturbation.

# B LPIPS Results

In Table 3 of the main paper, we used PSNR and SSIM to evaluate image similarity between the cover images and the corresponding steganographic images. In Table 6 we show the results of using (LPIPS) (Zhang et al., 2018), a newer perceptual image similarity metric, to evaluate image similarity between cover and steganographic images. We observe similar trends as with PSNR and SSIM.

| Dataset | Method | LPIPS ↓ | | | |
|---|---|---|---|---|---|
| | | 1 bit | 2 bits | 3 bits | 4 bits |
| Div2k | SteganoGAN | 0.13 | 0.13 | 0.14 | 0.14 |
| | LISO | 0.04 | 0.05 | 0.06 | 0.08 |
| | FNNS-D | 0.01 | 0.02 | 0.08 | 0.07 |
| | LISO+L-BFGS | 0.04 | 0.05 | 0.07 | 0.08 |
| CelebA | SteganoGAN | 0.15 | 0.15 | 0.14 | 0.15 |
| | LISO | 0.09 | 0.08 | 0.12 | 0.15 |
| | FNNS-D | 0.07 | 0.08 | 0.15 | 0.15 |
| | LISO+L-BFGS | 0.06 | 0.06 | 0.11 | 0.14 |
| MS COCO | SteganoGAN | 0.12 | 0.13 | 0.13 | 0.13 |
| | LISO | 0.04 | 0.05 | 0.09 | 0.14 |
| | FNNS-D | 0.01 | 0.01 | 0.05 | 0.05 |
| | LISO+L-BFGS | 0.07 | 0.06 | 0.09 | 0.15 |

Table 6: Steganographic image quality measured with the LPIPS metric (Zhang et al., 2018). Lower numbers indicate better image quality.

# C Error vs Iteration

Figure 6 shows how the recovery error rate decreases every iteration. As we see, the error rate monotonically decreases and this implies that LISO learns a good descent direction. We also see that 15-20 iterations are enough for the error rate to converge.

# D Loss for Different Optimization Methods

Figure 7 is analogous to figure 4 in the main paper. However, instead of seeing how the error rate decreases for different optimization methods, we see how the loss decreases. Note that the first data point we plot for each curve is after 1 iteration and we do so to improve the visualization. Similar to figure 4, we see that the loss for LISO optimization decreases much faster than any other method.

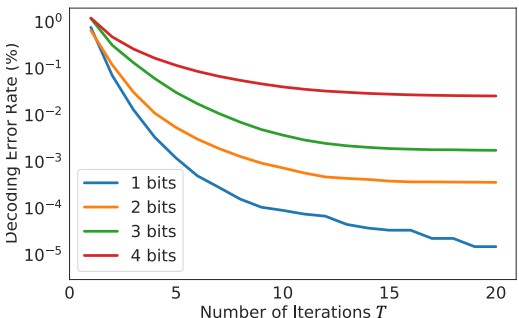 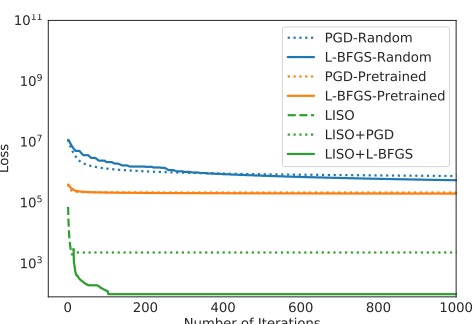

Figure 6: LISO recovery error rate with regard to number of iterations $T$ under different bit rates, evaluated on Div2k's validation set. Y-axis is shown in the log scale.

Figure 7: Loss-iteration curves for different optimization methods (at 4 bpp). The pretrained network is SteganoGAN.

## E    COMPARISON WITH CHAT-GAN

Tan et al. (2021) present steganography results for hiding 1-4 bpp messages in images from the COCO dataset using CHAT-GAN. We train LISO using different $\lambda$ weights to match the error rates of CHAT-GAN (as listed in their paper) as closely as possible and the results are presented in Table 7; we use $\lambda = 100$ for 1-2 bpp and $\lambda = 20$ for 3-4 bpp for training LISO. From the table, we see that the error rates and SSIM values are comparable for both methods, but CHAT-GAN obtains better PSNR numbers. This is likely due to the fact that CHAT-GAN has a specially designed feature attention module to hide bits imperceptibly. However, as evidenced by both the SSIM values and the qualitative images below, steganographic images from LISO visually look indistinguishable from the cover images despite having lower PSNR values (PSNR is a local statistic and it measures the absolute mean squared difference in pixel values and two images can look similar despite have a low PSNR). Moreover, note that the contributions of our paper are orthogonal to the contributions of Tan et al. (2021). We can also combine our iterative LISO method with the CHAT-GAN encoder network architecture (Tan et al., 2021) to take advantage of both methods and to perform steganography with higher payloads and harder images.

| Capacity | Method | Error (%) ↓ | PSNR ↑ | SSIM ↑ |
|----------|--------|-------------|--------|--------|
| 1 bpp | LISO | 0.34 | 38.02 | 0.99 |
| | CHAT-GAN | 0.93 | 46.42 | 0.99 |
| 2 bpp | LISO | 1.36 | 34.25 | 0.98 |
| | CHAT-GAN | 1.20 | 43.17 | 0.99 |
| 3 bpp | LISO | 0.60 | 30.06 | 0.93 |
| | CHAT-GAN | 3.82 | 41.84 | 0.99 |
| 4 bpp | LISO | 4.87 | 26.16 | 0.83 |
| | CHAT-GAN | 5.44 | 38.92 | 0.95 |

Table 7: Comparing the performance of LISO with CHAT-GAN

## F    COMPARISON OF DIFFERENT STEGANOGRAPHY METHODS

Table 8 compares LISO with both learned methods like SteganoGAN (Zhang et al., 2019) and CHAT-GAN (Tan et al., 2021), and optimization based methods methods like FNNS (Kishore et al., 2021). We see that LISO has a trained encoder like learned methods, but its learned encoder is iterative and emulates the optimization algorithm in optimization-based steganography methods.

| Method | Type | Encoder | Speed | Decoder |
|---|---|---|---|---|
| Learned Methods | Learned | Learned 1-step Network | Fast | Learned Network |
| FNNS-R (Kishore et al., 2021) | Optimization | L-BFGS / PGD | Slow | Random Network |
| FNNS-D (Kishore et al., 2021) | Optimization | L-BFGS / PGD | Slow | Fixed Pre-trained Network |
| LISO (ours) | Hybrid | Learned Iterative Network | Fast | Learned Network |

Table 8: A comparison of how different learned and optimization based steganography methods differ from each other.

## G  LISO RESULTS WITH MSE LOSS WEIGHT $\lambda = 10$ FOR HIGHER IMAGE QUALITY

If higher image quality is desired, a higher mse loss weight $\lambda$ can be used. Consequently, the image quality will be better but the error rate will be worse. We repeat the experiments in Table 1 of the main paper with mse loss weight $\lambda = 10$, and show the results in Table 9. As seen in the table, the error rate is slightly worse (it's still less than 1.5%), but the image quality is better. We can further increase the value of $\lambda$ to obtain even better image quality.

| Dataset | Method | Error Rate (%) ↓ | | | | PSNR ↑ | | | | SSIM ↑ | | | |
|---|---|---|---|---|---|---|---|---|---|---|---|---|---|
| | | 1 bit | 2 bits | 3 bits | 4 bits | 1 bit | 2 bits | 3 bits | 4 bits | 1 bit | 2 bits | 3 bits | 4 bits |
| Div2k | LISO | 5.1E-4 | 1.0E-2 | 6.5E-2 | 3.8E-1 | 36.40 | 36.62 | 32.85 | 30.88 | 0.95 | 0.95 | 0.91 | 0.87 |
| | LISO* | 0 | 0 | 3.8E-6 | 5.8E-2 | 34.86 | 35.49 | 32.41 | 29.45 | 0.92 | 0.93 | 0.91 | 0.82 |
| CelebA | LISO | 1.5E-3 | 6.4E-3 | 4.5E-2 | 1.1E+0 | 35.81 | 36.80 | 35.19 | 32.94 | 0.90 | 0.92 | 0.89 | 0.84 |
| | LISO* | 0 | 0 | 1.3E-6 | 8.4E-1 | 35.13 | 35.91 | 34.35 | 32.89 | 0.89 | 0.91 | 0.88 | 0.84 |
| MS COCO | LISO | 1.8E-3 | 7.6E-3 | 1.6E-1 | 1.4E+0 | 34.32 | 36.43 | 31.46 | 30.14 | 0.91 | 0.95 | 0.86 | 0.81 |
| | LISO* | 0 | 0 | 1.2E-6 | 3.2E+0 | 33.20 | 35.23 | 30.35 | 29.57 | 0.90 | 0.93 | 0.83 | 0.78 |

Table 9: Image steganography results of LISO with loss weight $\lambda = 10$. Note that LISO* is LISO+L-BFGS and was shortened due to space constraints.

## H  ADDITIONAL STEGANALYSIS

In addition to SiaStegNet (You et al., 2020), we attempt to avoid detection from two other state-of-the-art steganalysis systems: SRNet (Boroumand et al., 2018) and XuNet (Xu et al., 2016). Results are reported in Table 10. For most cases, the results are similar to SiaStegNet. However, with XuNet we see a very high detection accuracy in the "w/o defense" scenario; this is because XuNet uses handcrafted kernels in addition to using a convolutional network to detect steganographic images. Despite this, adding the loss XuNet allows us to evade detection from it (as seen from the "w/ defense" scenario). We can also achieve lower detection accuracy in the "w/o defense" scenario if a method like XuNet (with hand-crafted kernels) is used as a discriminator/critic during LISO training.

| Method | Error Rate (%) ↓ | | | | PSNR ↑ | | | | Detection Accuracy Rate (%) ↓ | | | |
|---|---|---|---|---|---|---|---|---|---|---|---|---|
| | 1 bit | 2 bits | 3 bits | 4 bits | 1 bit | 2 bits | 3 bits | 4 bits | 1 bit | 2 bits | 3 bits | 4 bits |
| SRNet (w/o defense) | 1E-04 | 3E-04 | 5E-03 | 4E-02 | 33.83 | 34.15 | 30.08 | 24.58 | 51 | 40 | 33 | 74 |
| SRNet (w/ defense) | 6E-04 | 1E-04 | 1E-03 | 2E-01 | 33.43 | 32.74 | 28.51 | 24.89 | 0 | 0 | 0 | 1 |
| XuNet (w/o defense) | 1E-04 | 3E-04 | 5E-03 | 4E-02 | 33.83 | 34.15 | 30.08 | 24.58 | 100 | 98 | 100 | 100 |
| XuNet (w/ defense) | 2E-04 | 3E-03 | 1E-02 | 4E-02 | 34.16 | 32.98 | 28.40 | 25.31 | 2 | 2 | 42 | 100 |

Table 10: Steganalysis results with SRNet and XuNet. All experimental configurations follow Table 3 in main paper.

## I  EXPERIMENT RESULTS WITH HIGHER PAYLOADS

Extending the results from the main paper, in Table 11, we evaluate LISO with 5-6 bits encoded per pixel. We see that we are able to get low error rates ($< 3\%$) even for messages hiding 5-6bpp of information. However, the image quality is not very good; as described in Appendix G we can get images with better image quality but a slightly worse error rate if we increase the weight on the MSE loss term.

| Dataset | Method | Error Rate (%) ↓ | | PSNR ↑ | | SSIM ↑ | |
|---------|--------|-------|-------|-------|-------|-------|-------|
| | | 5 bits | 6 bits | 5 bits | 6 bits | 5 bits | 6 bits |
| Div2k | SteganoGAN | 31.44 | 35.35 | 20.05 | 20.34 | 0.79 | 0.80 |
| | LISO | 2.37 | 9.08 | 22.58 | 23.29 | 0.53 | 0.56 |
| | FNNS-D | 18.12 | 19.67 | 12.34 | 12.3 | 0.13 | 0.14 |
| | LISO+L-BFGS | 2.13 | 9.17 | 20.86 | 23.24 | 0.44 | 0.55 |
| CelebA | SteganoGAN | 32.15 | 31.16 | 19.51 | 21.82 | 0.74 | 0.79 |
| | LISO | 0.48 | 8.73 | 24.88 | 25.15 | 0.48 | 0.47 |
| | FNNS-D | 15.3 | 18.41 | 12.94 | 12.99 | 0.07 | 0.07 |
| | LISO+L-BFGS | 0.16 | 8.31 | 24.68 | 24.69 | 0.46 | 0.45 |
| MS COCO | SteganoGAN | 33.20 | 35.67 | 22.74 | 23.07 | 0.86 | 0.85 |
| | LISO | 1.09 | 8.66 | 21.32 | 22.63 | 0.36 | 0.45 |
| | FNNS-D | 16.41 | 17.44 | 15.63 | 15.78 | 0.19 | 0.20 |
| | LISO+L-BFGS | 1.52 | 8.50 | 21.08 | 22.55 | 0.35 | 0.45 |

Table 11: Image steganography results with 5-6 bits encoded per pixel.

## J    ADDITIONAL QUALITATIVE RESULTS

A few qualitative examples of steganographic images were presented in the main paper. Additional steganographic images produced by LISO with 1-4bpp of hidden information are shown in Figure 8. Stenographic images obtained with an increased mse weight $\lambda = 10$ are shown in Figure 9.

As explained in the paper, we can use L-BFGS with LISO to further reduce the error rate obtained from LISO. Figure 10 shows steganographic images obtained from LISO+L-BFGS and we see that there is no difference in visual quality even after the additional optimization.

In section 5 of the main paper we show that LISO can avoid being detected by SiaStegNet (You et al., 2020) by using the gradient from SiaStegNet in LISO's optimization network. In Figure 11, we show sample images from the Div2k dataset that are generated in this way to avoid SiaStegNet detection. Again we see no noticeable difference in visual quality.

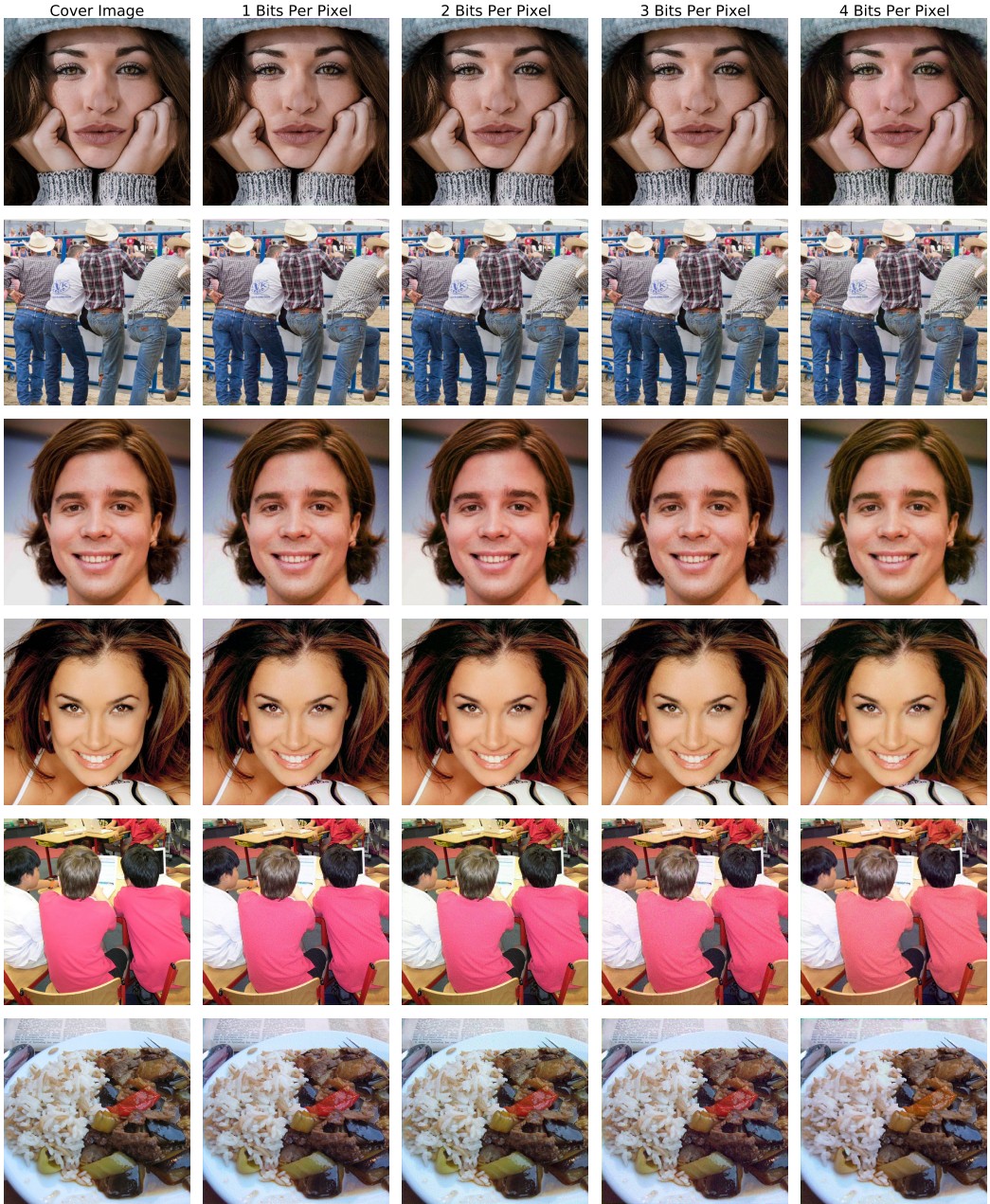

Figure 8: Cover images with corresponding steganographic images under different payloads. The first 2 images are from Div2k, the following 2 are from CelebA, and the last 2 are from MS-COCO.

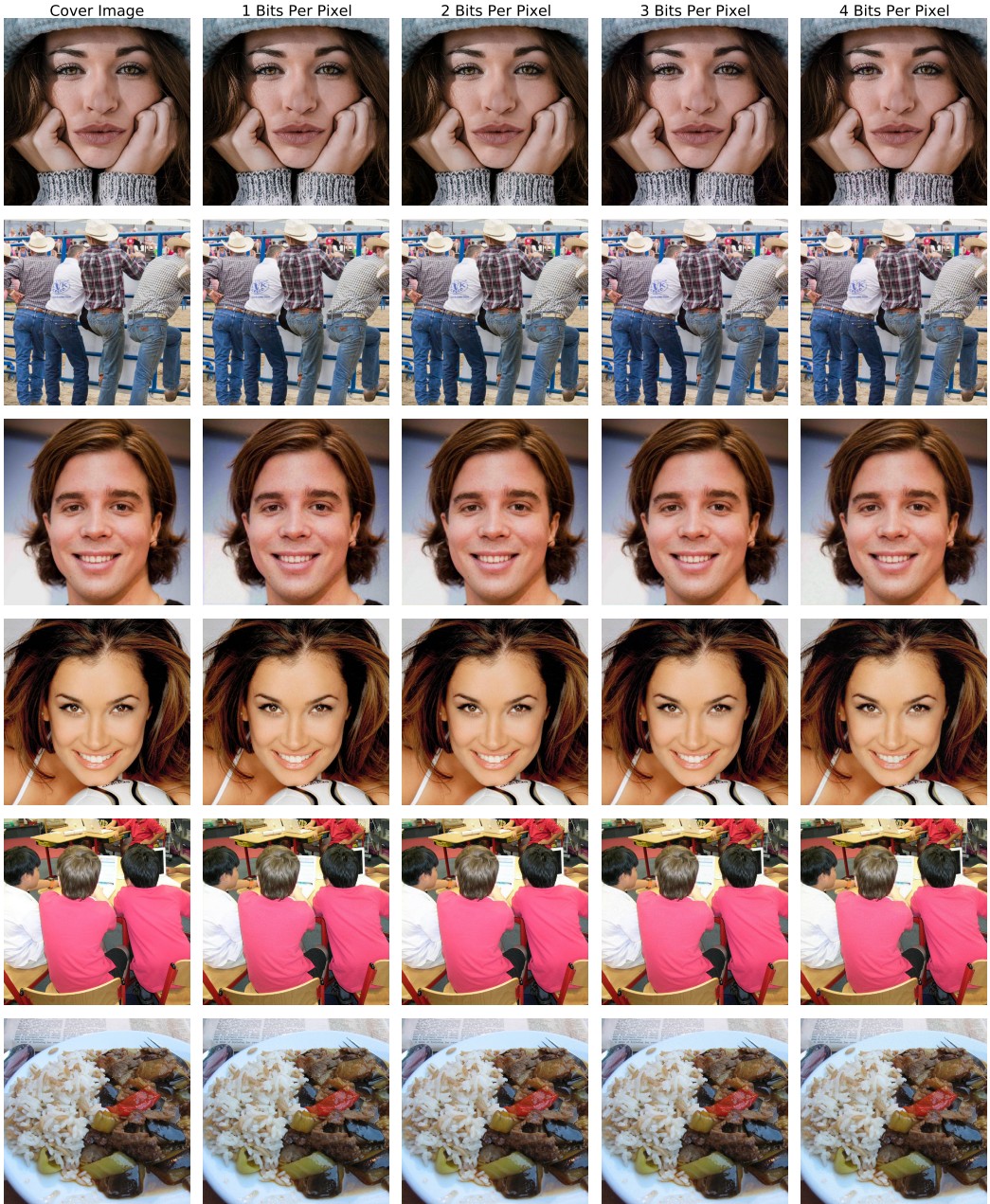

Figure 9: Cover images with corresponding steganographic images under different payloads trained with $\lambda = 10$.

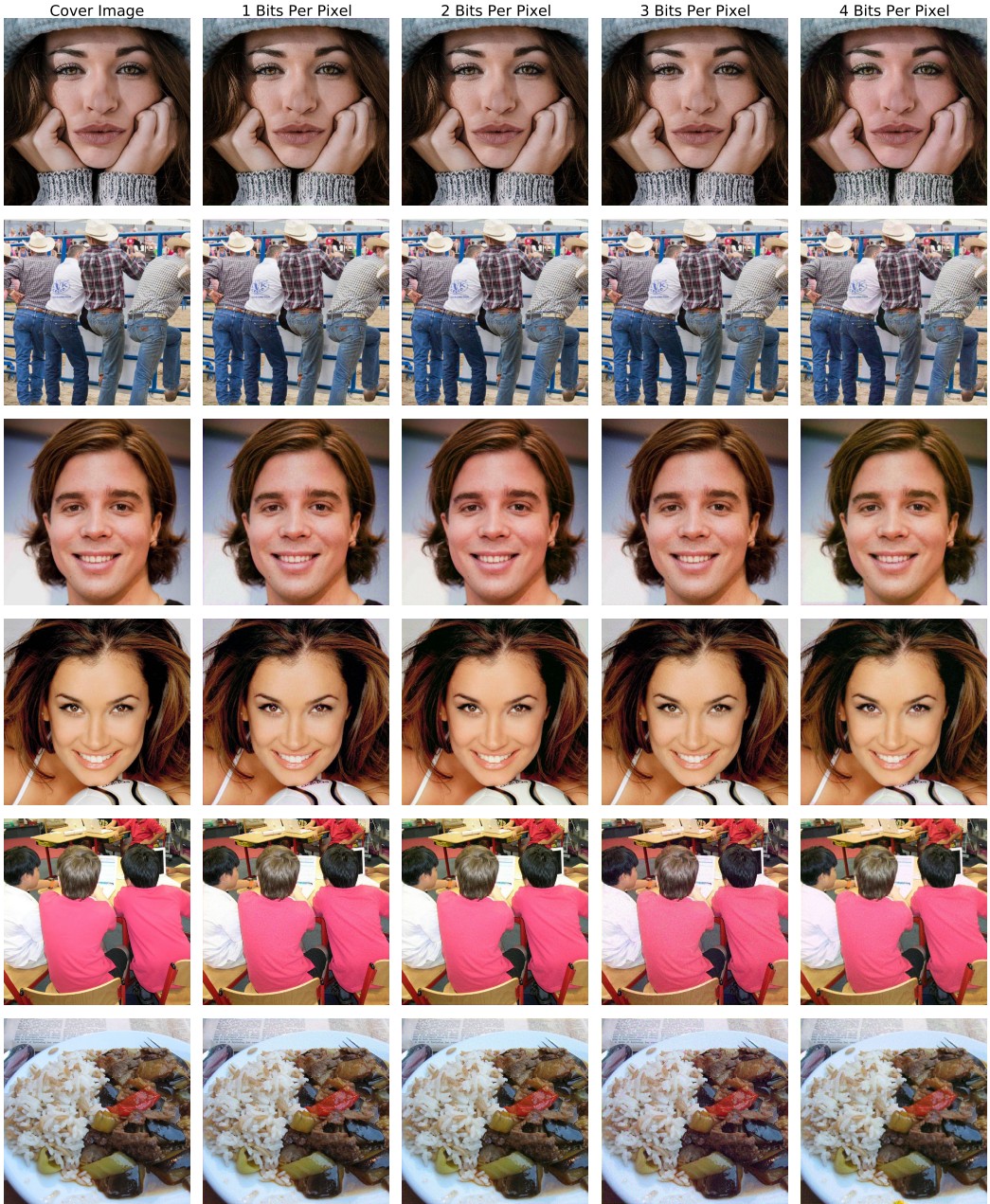

Figure 10: Cover images with corresponding steganographic images under different payloads using LISO+L-BFGS.

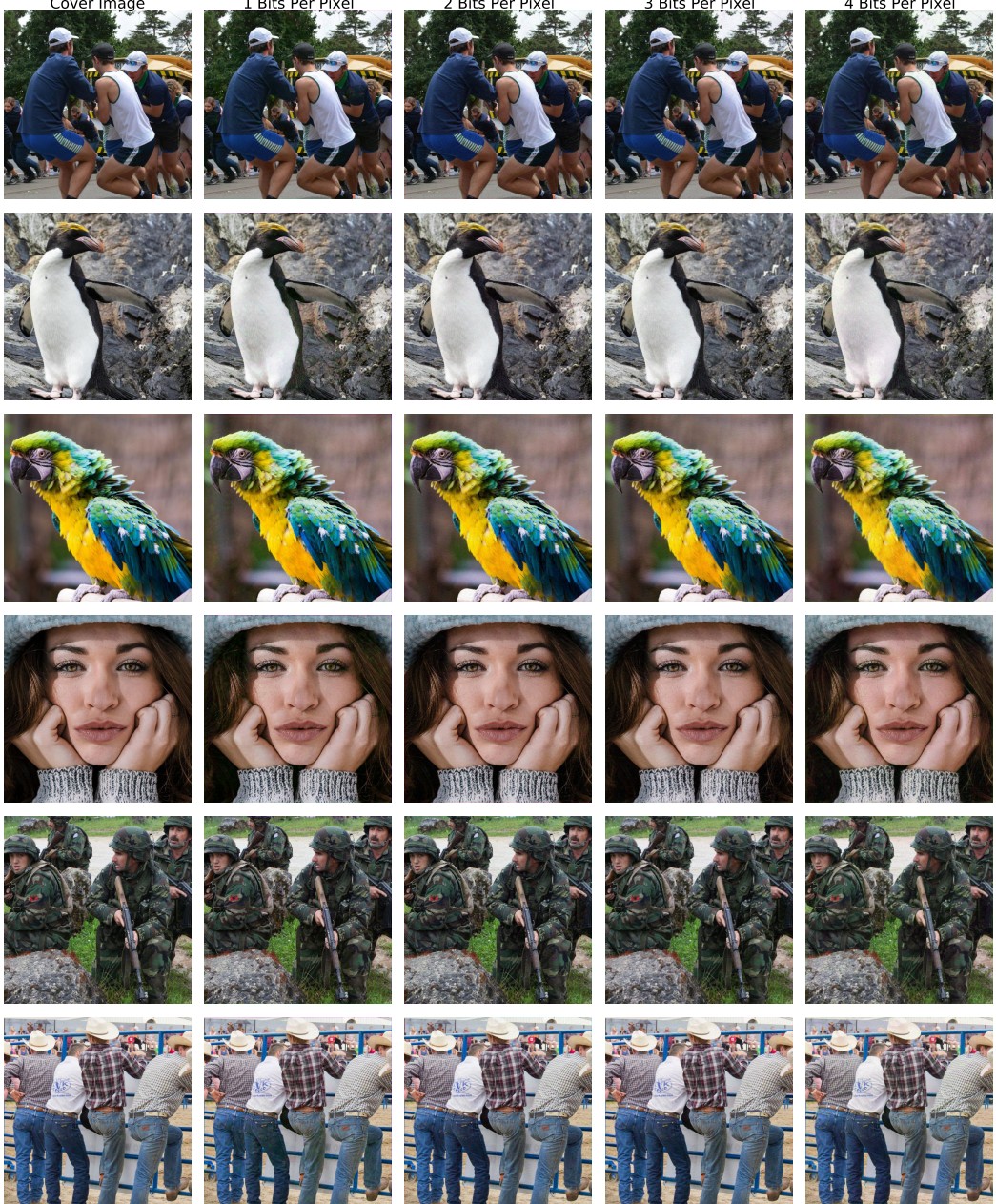

Figure 11: Steganographic produced by LISO with additional detection loss. These images can all avoid detection by SiaStegNet.

