# OpenReview forum: "Learning Iterative Neural Optimizers for Image Steganography"
_ICLR.cc/2023/Conference — ICLR 2023 poster_

### Official Review · Reviewer_3A9y · 2022-10-20

**Confidence:** 4
**Correctness:** 3
**Technical Novelty And Significance:** 3
**Empirical Novelty And Significance:** 3
**Recommendation:** 6

**Clarity, Quality, Novelty And Reproducibility:**

## Clarity
This work clearly written and technically sound to the best of my judgment.

## Quality
The paper quality is sufficient in my opinion.

## Novelty
In my judgment, this work holds novelty in terms of the proposed approach and the accompanying analysis and results.

## Reproducibility
Since the code is not provided, it might be difficult to reproduce this work.

**Strength And Weaknesses:**

## Strengths
(+) Deep-learning-based steganography approaches are an active research field, gaining more and more popularity. This work is one of a few, that investigates the optimization-based approach.
(+) This work presents convincing results, with a relatively low error rate and average time (outperforming FNNS).
(+) I highly appreciate the analysis regarding steganalysis, runtime, and JPEG compression.

## Weaknesses
(-) I might have missed it, but I would be interested regarding the hidden messages. Were the bit string messages chosen randomly? I am also interested if the authors encountered “adversarial” bit strings, meaning bit strings that are difficult to encode.
(-) I appreciate the evaluation in Appendix I. What are the limitations of the algorithm regarding the length of bit strings? Or in other words, at what payload lengths, does the algorithm not produce sufficient results?
(-) I am curious about a study regarding the cover image. Are there cover images, for which it is easier or more difficult to encode messages, e.g. random noise, black-and-white images, or single-colored images?
(-) The authors strongly argue that “the steganographic image remains on the natural image manifold with the help of a critic”. This argument is repeated multiple times throughout the paper, but no concrete evidence or proof is provided. Hence, this claim remains an assumption.
(-) Will the code be provided?


**Summary Of The Paper:**

This work proposes Learned Iterative Steganography Optimization (LISO), a steganography approach that leverages deep neural networks and iterative optimization to hide a secret message in an image. LISO is faster than previous optimization-based approaches and has low message recovery error.

**Summary Of The Review:**

The presented LISO is a novel steganography approach that outperforms previous optimization-based steganography approaches. The authors provide convincing results and analysis regarding their approach. From my judgment, this work holds sufficient novelty for a subgroup in the ML community.

---

> ### Author Response · Authors · 2022-11-11
> **Response to Reviewer 3A9y**
>
> Thank you for your detailed feedback. Below are the answers to your questions.
>
> * The bit string messages were chosen randomly and each bit follows an independent bernoulli distribution with p=0.5. We believe this is very close to the true setting, because in practice messages are typically compressed (using zip) or encrypted, and hence their bit strings are close to uniformly random in distribution.  We observed that hiding a pathological all-0s or all-1s bit-string is a more difficult optimization problem, but in practice such strings seem contrived for the reason mentioned above.
> * As shown in the appendix, even though we obtain low error rates (2-8%) for 5-6 bpp, the image quality is not great (about 23 PSNR). At 5-6bpp, we can obtain results with better image quality by increasing the weight lambda on the mse loss but the error rate will also be slightly higher. We haven’t pushed it higher than 6bpp but would expect the results to deteriorate as the optimization problem will be even harder.
> * We do observe that single-colored and low textured images are harder to encode messages in. This is to be expected as even small perturbations appear to be more visible in low textured regions. Thanks for bringing this up, we will add some discussion in that respect to the final version.
> * Prompted by your question, we ran an additional ablation in which we trained LISO networks without a critic, and found the image quality to be considerably worse. This effect was especially pronounced for higher bit rates, where the perturbation is larger and the image “travels” further from the original cover image. Thanks for the suggestion, we will add this ablation to the appendix.
> * Yes, we will release our code on GitHub with the camera ready version of the paper.
>
> Let us know if there are any other concerns or questions we can respond to!

---

### Official Review · Reviewer_PqKU · 2022-10-23

**Confidence:** 4
**Correctness:** 3
**Technical Novelty And Significance:** 3
**Empirical Novelty And Significance:** 3
**Recommendation:** 6

**Clarity, Quality, Novelty And Reproducibility:**

The paper is overall very clear and presents novel and interesting ideas on the problem of steganography.
The method presented by this paper seems to be reproducible.


**Details Of Ethics Concerns:**

There are no ethical issues to be addressed.

**Strength And Weaknesses:**


Strengths:

1) the paper is well written and it is very clear
2) the novel ideas of the paper explores two separate previous approaches to the problem, namely the neural learning and the iterative optimisation. The method presented by the paper is able to jointly explore these two approaches by creating a learning process for the optimisation problem, able to learn the gradient descent direction.
3) the experiments are thorough and show that the method outperforms the state-of-the-art

Weaknesses:
1) when comparing the method and the cited methods of the state-of-the-art, the authors usually state the recovery error rate of the latter ones, by 5-25% for 4bpp, and state that the presented method can achieve 0% recovery error rate for up to 3bpp. In my opinion, it would be more fair and clear to compare in the text both the presented method and SOTA in both 3bpp and 4bpp. notice that the results are stated in Table 1, but the text is not so clear mentioning it.
2) the paper does not explores sufficiently the resulting image quality in terms of being natural (indirectly measured by PNSR and SSIM). The results of Table 1 show that the proposed method does not always achieve better results in terms of image quality, so I suggest a better discussion in terms of image quality and maybe including some subjective evaluation of images by humans.
3) In section 4, in the Training paragraph, it is not very clear whether the end-to-end training is image-specific or uses several different images.
4) There are a few typos. I suggest a carefully revision of the text.


**Summary Of The Paper:**

This paper presents a new method for image steganography based on an iterative optimisation method (gradient-descendent) able to achieve zero recovery error rate for encodings up to 3bpp. The method is able to keep steganography output images close to (or on) the manifold created by natural images. The experimental results show that the paper outperforms the state-of-the-art in terms of recovery (decoding) error rate and in terms of computational cost.

**Summary Of The Review:**

The paper is interesting and well written. It would benefit of some improvements in terms of comparison of the recovery error rate and, mainly, in terms of resulting image quality.

---

> ### Author Response · Authors · 2022-11-11
> **Response to Reviewer PqKU**
>
> Thank you for your constructive suggestions. Here are a few additional clarifications:
>
> * We agree that the text would be clearer were we to present LISO and SOTA numbers for 3bpp and 4bpp. Thank you for pointing this out; we will edit the text accordingly.
> * Qualitatively, the image quality is similar for LISO and FNNS. As stated by both you and reviewer UMSE, PSNR and SSIM are more closely aligned with distortion than perception/realism and may not be the best measures of image quality. We will include additional images in the appendix, so that readers can verify the image quality for images produced by LISO. We will also add additional discussion.
> * The training is not image specific — the LISO network is trained end-to-end on a dataset of images. Therefore, unlike FNNS, LISO can leverage information learned from optimizing multiple images. Thanks for drawing our attention to section 4; we will clarify the writing in the final version.
> * Thanks for the heads up; we’ll check and ensure there are no typos in the camera-ready version.
>
> Please let us know if you have any additional questions.

---

### Official Review · Reviewer_UMSE · 2022-10-25

**Confidence:** 3
**Correctness:** 4
**Technical Novelty And Significance:** 3
**Empirical Novelty And Significance:** 3
**Recommendation:** 8

**Clarity, Quality, Novelty And Reproducibility:**

The paper is clear and the presentation and empirical evaluation are of high quality. I have not previously seen GRU (or other RNNs) used for iterative optimization so I believe this approach is novel.

Although the paper is conceptually clear, releasing code would improve reproducibility considerably since there may be many details that effect performance in how the GRU-based optimization and GAN-loss are implemented.

**Strength And Weaknesses:**

The primary strength of the paper is the interesting approach and solid empirical results. The approach is based on iterative optimization using GRU where each iteration takes the previous estimate of the perturbation and a gradient, and the hidden state allows the method to minimize the loss more quickly and more effectively than methods that only take a gradient (see Fig 3).

The empirical evaluation involves three components: the error rate in recovering the hidden message, the image quality after embedding the message, and runtime. Table 1 shows that LISO (coupled with L-BFGS) outperforms SteganoGAN by a significant margin. Compared to FNNS-D, both methods reach zero rates (or very close to it) up to 3 bits per pixel (bpp) but only LISO has near-zero error rates at 4bpp. On the other hand, FNNS-D does have higher PSNR and SSIM for two out of the three data sets (MS-COCO and CelebA but not Div2k). It's not clear to me how big of a problem this is, though, since LISO uses a GAN-like loss to keep perturbed images near the manifold of natural images. I would expect this loss to make the images look more realistic at the expense of fidelity (this is often discussed as high perceptual quality but also high distortion). Since PSNR and SSIM are more closely aligned with distortion than perception/realism, they may not be good measures of image quality, or, at least, they're at odds with the GAN-like loss used by LISO (see work on the perception-distortion tradeoff by Blau & Michaeli for more information).

Furthermore, LISO runs much more quickly than FNNS-D and LISO+L-BFGS has a significant speed advantage over FNNS-D too.

In practice, the main weakness of LISO is that performance drops with lossy compression, however LISO-JPEG (a variant of LISO that incorporates JPEG compression into the training procedure) fairs better than the baseline methods. Since natural images are almost always compressed using a lossy codec, the main results ("LISO-PNG" that uses *lossless* compression) has relatively little real-world impact.


**Summary Of The Paper:**

This paper presents a method called Learned Iterative Steganography Optimization (LISO) that performs image steganography, i.e. embedding arbitrary information in image data (not metadata) in a way that is difficult to see or detect through statistical analysis. LISO uses iterative optimization based on a gated recurrent unit (GRU) to determine how to perturb image pixels to minimize visibility and maximize recovery of an arbitrary message. The authors show that compared to existing steganography methods, LISO is fast (Table 4), leads to low error rates in recovering the hidden message (Table 1), and can be adapted to more resilient to JPEG compression and more resistant to automated detection.


**Summary Of The Review:**

I'm recommending that this paper be accepted based on the interesting use of GRU for fast (low iteration count) optimization, and the strong empirical results presented.

Initially, I wasn't sure that steganography applications were a good fit for ICLR, at least without further analysis of the actual representation that's learned, but the main baseline (FNNS) was published at ICLR 2021. This work appears to be an interesting and significant improvement over FNNS.

---

> ### Author Response · Authors · 2022-11-11
> **Response to Reviewer UMSE**
>
> Thanks for your thorough review and encouraging response. Although the LISO performance is admittedly still low for lossy compression, it is fair to say that this is a very hard problem for all high bit rate methods. LISO is setting a new state-of-the-art with respect to decoding error rate and we hope that future research will close the performance gap between compressed and uncompressed images further. We believe our approach can be extended to find perturbations in a low-frequency space (such that they are resistant to compression), although this remains to be validated, and would be a significant extension of the present work. We will definitely release our code upon publication.

---

### Official Review · Reviewer_cBgs · 2022-10-28

**Confidence:** 3
**Correctness:** 4
**Technical Novelty And Significance:** 1
**Empirical Novelty And Significance:** Not applicable
**Recommendation:** 8

**Clarity, Quality, Novelty And Reproducibility:**

Clarity: Good.
Quality: Good.
Novelty: Good.
Reproducibility: The authors have not claimed to release code.

**Strength And Weaknesses:**

Strengths:
* The paper is well-written and easy to follow.
* The proposed method is intuitive and novel. The rationale behind applying learnable blocks iteratively is explained clearly.
* The experiment results are strong and convincing.

Weaknesses.
Minor points:
* Figure 1 could include a scaled version of the diff if space allows.
* Figure 4 CelebA sample has a very visible color shift a 1bit per pixel. Given the relatively high average PSNR reported, are the shown results correct or an outlier in the dataset? From the supplementary material samples, other images do not show this color shift.


**Summary Of The Paper:**

This paper proposes a method which combines the benefits of iterative optimization and a learned encoder-decoder for the task of image steganography. Similar to the style of LISTA, the paper learns individual optimization blocks in an E2E fashion while applying the blocks iteratively at inference time. Experiment results show superior performance in terms of error rate and PSNR compared to previous baselines.



**Summary Of The Review:**

Overall I believe this paper has enough novelty and strong enough experiment results to recommend acceptance.

---

> ### Author Response · Authors · 2022-11-11
> **Response to Reviewer cBgs**
>
> Thanks for your encouraging review. We respond to your concerns below.
>
> * The first row in figure 1 shows the perturbation, which is the difference between the original (cover) image and the perturbed (steganographic) image. It may not have been clear that this is the diff; we will clarify it in the final version.
> * The CelebA example shown in figure 4 is indeed an outlier. A majority of the images do not show a color shift (as seen in the examples from the appendix). We did observe a small fraction of images which show a color shift as seen in figure 4, however only if the LISO model was trained on CelebA. We speculate that this is due to the lack of diversity in the training data. When we use a model trained on the Div2k dataset to perform steganography with the CelebA images, we find that there is no longer a color shift (a similar observation was made by [1] for image transformations).
>
> Let us know if further classification is needed.
>
> [1] Upchurch, P., Gardner, J., Pleiss, G., Pless, R., Snavely, N., Bala, K., & Weinberger, K. (2017). Deep feature interpolation for image content changes. In Proceedings of the IEEE conference on computer vision and pattern recognition (pp. 7064-7073).

---

### Decision · Program_Chairs · 2023-01-20

**Decision:**

Accept: poster

**Justification For Why Not Higher Score:**

Two of the four reviewers were a bit less enthusiastic, with scores leaning accept.  All four reviewers found minor weaknesses in the paper.

**Justification For Why Not Lower Score:**

All four reviewers ultimately are supporting of the paper being published, so there is consensus on this being accepted to the conference.

**Metareview: Summary, Strengths And Weaknesses:**

Thanks for your submission to ICLR.  The reviewers were generally positive about this paper, with all four reviewers recommending to accept.  On the positive side, the reviewers liked the clarity of the paper/writing, the novelty of ideas, and the thorough experimentation.  They all mentioned weaknesses as well, but these were all minor.

**Note From Pc:**

if the above contains the word "oral" or "spotlight" please see: "oral" presentation means -> notable-top-5% and "spotlight" means -> notable-top-25%. As stated in our emails, we are disassociating presentation type from AC recommendations